# Effectiveness of a Comprehensive Mental Health Literacy Educational Programme for Junior High School Students: A Randomised Controlled Trial Examining Changes in Their Knowledge, Attitudes, and Behaviour

**DOI:** 10.3390/jpm12081281

**Published:** 2022-08-04

**Authors:** Ryoichi Mori, Takashi Uchino, Masafumi Mizuno, Taiju Yamaguchi, Naoyuki Katagiri, Takahiro Nemoto

**Affiliations:** 1Department of Neuropsychiatry, Toho University Graduate School of Medicine, 5-21-16 Omori-nishi, Ota-ku, Tokyo 143-8540, Japan; 2Department of Physical Education, Tokai University, 4-1-1 Kitakaname, Hiratsuka 259-1292, Japan; 3Department of Neuropsychiatry, Toho University Faculty of Medicine, 6-11-1 Omori-nishi, Ota-ku, Tokyo 143-8541, Japan; 4Tokyo Metropolitan Matsuzawa Hospital, 2-1-1 Kamikitazawa, Setagaya-ku, Tokyo 156-0057, Japan

**Keywords:** health behaviour, health education, help-seeking, mental health literacy, mental illness, stigma

## Abstract

Background: This study evaluated the effectiveness of a comprehensive mental health literacy (MHL) educational programme known as “Sanita” for improving junior high school students’ knowledge of mental illness, attitudes towards people with mental health problems, and help-seeking behaviour. Methods: A randomised controlled trial with a parallel-group design was conducted. A total of 125 students (51 in the intervention group and 74 in the control group) received three 50-min classes and completed self-report questionnaires (Mental Illness and Disorder Understanding Scale, MIDUS; Reported and Intended Behaviour Scale, RIBS-J; and an original questionnaire investigating help-seeking behaviour) before and after the programme and three months later. Results: Regarding MIDUS, the post-test and 3-month follow-up test results showed a significant main effect of time-by-group interactions in a linear mixed model. Regarding RIBS-J, the post-test results showed a significant main effect of time-by-group interactions; however, the 3-month follow-up test showed no significant effect. No significant effects of time-by-group interactions were seen in the post-test and 3-month follow-up test results for help-seeking behaviour in a logistic regression-mixed model. Conclusions: The Sanita MHL educational programme was longitudinally effective at improving junior high school students’ knowledge of mental illness, although improvements in attitudes and help-seeking behaviour were insufficient.

## 1. Introduction

The social burden associated with mental illness is enormous, and the impact on young people is particularly severe, as around three quarters of all mental illnesses pre-sent by the age of 25 years [1,2]. There is a worldwide movement towards early intervention for the prevention of and recovery from mental illness, and evidence of the effectiveness of early intervention is accumulating. For example, early intervention in patients with psychotic disorders has been reported to improve the long-term prognosis [3]. Furthermore, attempts have been made to provide support for individuals with an at-risk mental state before the onset of overt illness [4,5]. However, the duration of untreated psychosis (DUP) is often very long for patients with psychotic disorders [6,7], and large epidemiological studies have shown that many people who meet the diagnostic criteria for depression and neurosis do not receive treatment [8]. These results suggest that young people who develop mental illness are not receiving treatment early enough.

One of the main reasons for this serious situation is the existence of a number of barriers that hinder youth from seeking help from others when they present with mental health problems [9,10]. Major barriers to help-seeking behaviour include a lack of knowledge of mental illness, stigmatising attitudes, and poor physical access to services [11]. From a health education perspective, the classic Knowledge–Attitude–Behaviour (KAB) model can be applied to promote changes in help-seeking behaviour; this model stresses the need to acquire appropriate knowledge of mental illness, and to develop attitudes that reduce stigma and enable appropriate coping with mental health problems [12,13,14,15]. Subsequently, educational models, such as the PRECEDE-PROCEED model [16], have been developed even further, but the acquisition of appropriate knowledge and attitudes has remained a necessary condition for behavioural change [17]. In other words, the acquisition of mental health literacy (MHL) is essential to promote behavioural change in relation to help-seeking behaviour. MHL is defined as “understanding how to obtain and maintain positive mental health; understanding mental disorders and their treatments; decreasing stigma related to mental disorders; and, enhancing help-seeking efficacy (knowing when and where to seek help and developing competencies designed to improve one’s mental health care and self-management capabilities)” [18]. A variety of educational programmes have been developed around the world to improve MHL for young people, mainly in school settings [19,20,21].

To date, most studies of the effectiveness of MHL educational programmes have examined the outcomes of increased knowledge and attitudes about mental illness [22]. Only a few studies have longitudinally investigated whether implementing these pro-grammes has resulted in changes in the actual help-seeking behaviour of participants, which is one of the ultimate goals of their implementation [23]. These results have been inconsistent, and further work is needed. Furthermore, a recent meta-analysis reported that the peak age of the onset of overall mental illness in the general population is 14.5 years [24]; therefore, ideally, by this age at the latest, appropriate changes in help-seeking behaviour should have been achieved. Unfortunately, most of the existing programmes are targeted at high-school-aged youths, and there are limited practices targeting the age group before the age group during which the peak onset of mental illness occurs [22,25].

In view of the above, the establishment of a MHL programme for junior high school students that will enable them to acquire appropriate knowledge of not only mental ill-ness and attitudes towards stigma, but also help-seeking pathways, is urgently needed. The present study focused on the Japanese MHL educational program known as “Sanita” [26,27]. A unique feature of the Sanita programme is that, unlike the short lectures used in many other programmes, the Sanita programme is designed to be as comprehensive as possible, and can be used by any teacher in a real-world school setting. In particular, the lectures include animations that are highly acceptable to young people, and video interviews with peers that make mental illness accessible to everyone, with the aim of developing appropriate knowledge and attitudes. In addition, exercises and group discussions to develop a specific help-seeking behaviour plan in the event of a mental health problem are used to promote behavioural change. In the present study, we aimed to examine the effectiveness of this programme. We conducted a randomised con-trolled trial (RCT) that investigated changes in the participants’ knowledge of mental illness, attitudes towards people with mental health problems, and help-seeking behaviour.

## 2. Materials and Methods

### 2.1. Participants

The subjects consisted of 156 first-year students attending a public junior high school (aged 12–13 years; Grade 7; Year 8) in Tochigi, Japan. The choice of school was based on the availability of study cooperation and the feasibility of the intervention. The exclusion criteria were those who could not read or write Japanese at a level equivalent to that of a Japanese primary school graduate. Participants in the study provided written consent from both the subjects themselves and their parent or guardian. The study protocol was approved by the Ethics Committee of the Faculty of Medicine, Toho University (approval number: A22021_A21054_A19088). This study was performed in accordance with the latest version of the Declaration of Helsinki.

### 2.2. Study Design

We conducted an individual-level RCT with a parallel-group design. The randomly allocated intervention group received three hours of the Sanita MHL educational programme, whereas the control group received three hours of regular health classes given in junior high schools in Japan. Each group received one hour of class per week. Random allocation was made on a class-by-class basis by a research assistant who was not involved in the intervention, evaluation, or data analysis. Assessments were performed at baseline, after the completion of the classes, and 3 months after the classes. The survey was conducted using an anonymous self-administered questionnaire. This study was described as an assessor-blinded RCT in its registration. The study protocol was registered at the University Hospital Medical Information Network Clinical Trial Registration before the start of the study (trial number: UMIN000041397); no methodological changes were required after registration. This study was performed in accordance with the latest version of the Declaration of Helsinki.

### 2.3. Interventions

We used the Sanita MHL educational program [26,27]. The educational resources included short-story animated films, filmed social contact, and educators’ manuals, which are freely available through the internet. These resources were originally designed for high school students (grades 10–12) and were intended to be delivered by health and physical education teachers. In the present study, the contents were reorganised into three 50-min classes. These were given once a week for three weeks. Table 1 summarises the flow and content of these classes.

The control group received three 50-min classes on “Healthy Living and Disease Prevention”. These control classes represented the current general contents of health education in Japanese junior high schools, as outlined in the Courses of Study [28]. The contents were aimed at promoting a general understanding of the mechanisms of health and the development of disease in general. Specific MHL content was not included.

### 2.4. Measures

The Mental Illness and Disorder Understanding Scale (MIDUS) and the Japanese version of the Reported and Intended Behaviour Scale (RIBS-J) were used. In addition, a new self-administered questionnaire was developed to investigate whether the participants engaged in help-seeking behaviour. These questionnaires were administered at all three time-points: baseline, post-classes, and follow-up.

The MIDUS assesses the practically useful knowledge of mental illness. It consists of 15 items on a five-point Likert scale (range of 0 to 60, with a lower score representing more practically useful). The factorial validity and internal consistency have been confirmed [29]. It consists of three subscales: treatability of illness (e.g., “Mental illness is treatable”), efficacy of medication (e.g., “Medication is effective for improving symptoms”), and social recognition of illness (e.g., “Mental illnesses are very common”) [29]. We used a total score in this study.

The RIBS-J assesses behaviour and attitudes towards people with mental health problems, and consists of two parts: the RIBS-J past, which examines past experiences, and the RIBS-J future, which examines intentions about future behaviour (i.e., attitudes). In this study, the RIBS-J future was used. This part consists of 15 items on a five-point Likert scale (range of 4 to 20, with a higher score indicating a more positive intention). Both the original and Japanese versions have good validity and reliability [30,31]. An example of a question is: “In the future, I would be willing to live with someone with a mental health problem”.

A self-administered questionnaire consisting of four questions was newly developed to investigate whether the participants had engaged in any help-seeking-related behaviour in the previous three months. The four questions were as follows: (I) In the last three months, have you talked to your family about mental illness? (II) In the last three months, have you visited a website about mental illness? (III) In the last three months, have you talked to a friend about mental illness? (IV) In the last three months, have you consulted someone about mental illness? The choices were “yes”, “no”, or “don’t know”, and those who answered “yes” were identified as having engaged in help-seeking behaviour.

### 2.5. Data Analysis

Participants who properly completed all the assessments at all three time-points (baseline, post-classes, and follow-up) were included in the analyses. To test the differences in the score and proportion of correct or desirable answers to each measure over three time-points, we applied a random effect of intercept and slope in a mixed-effect model with a full maximum likelihood estimation. A linear mixed-model and a logistic regression-mixed model were used to analyse continuous variables (MIDUS and RIBS-J future) and categorical variables (behavioural measures), respectively. Statistical significance was set at 5% (*p* < 0.05). Statistical analyses were conducted using SPSS, version 26.0.

## 3. Results

The subjects were 156 students from one junior high school. Of these, 125 consented to participate in the study and were randomly allocated to the intervention group (n = 51) or the control group (n = 74). After excluding those who were absent from the classes or did not complete the assessments, the final number of participants included in the analysis was 116 (50 in the intervention group and 66 in the control group) (Figure 1). There were no differences in the demographic characteristics or outcome measure scores at baseline between the groups (Table 2).

There was a significant main effect of time and time-by-group interaction in the MIDUS post-test (B (95% CI), *p* value: main effect of time = −8.40 (−10.07, −6.73), *p* < 0.001; time-by-group interaction = 7.36 (5.14, 9.57), *p* < 0.001) and the 3-month follow-up (B (95% CI), *p* value: main effect of time = −6.14 (−7.81, −4.47), *p* < 0.001; time-by-group interaction = 4.19 (1.97, 6.41), *p* < 0.001) (Table 3).

There was a significant main effect of time and time-by-group interaction in the RIBS-J future post-test (B (95% CI), *p* value: main effect of time = 2.20 (1.45, 2.96), *p* < 0.001; time-by-group interaction = −1.85 (−2.85, −0.85), *p* < 0.001). There was a significant effect of the main effect of time in the 3-month follow-up for RIBS-J; however, the effect of time-by-group interaction was not significant (the 3-month follow-up (B [95% CI], *p* value: main effect of time = 1.84 [1.85, 2.60], *p* < 0.001; time-by-group interaction = −0.72 [−1.72, 0.28], *p* = 0.158)) (Table 3).

There were no significant main effects of time or time-by-group interaction in either the post-test or the 3-month follow-up questionnaire results for help-seeking behaviour (i.e., “consulting someone”). Help-seeking behaviour, such as “talking to family”, “visiting a website”, and “talking to a friend”, were significantly associated with the main effects of time in the post-test, although the effects of time-by-group interaction were not significant (B (95% CI), *p* value: main effect of time “talking to family” = 0.39 (0.19, 0.78), *p* = 0.008, “visiting a website” = 0.27 (0.13, 0.53), *p* < 0.001, “talking to a friend” = 0.24 (0.11, 0.52), *p* < 0.001). There were no significant main effects of time or time-by-group interaction in the 3-month follow-up questionnaire results for help-seeking behaviour (Table 4).

## 4. Discussion

This study is a multidimensional examination of the effectiveness of the Sanita MHL educational programme, which is designed to be as comprehensive as possible, and can be implemented by any teacher in a real-world school setting. To the best of our knowledge, this is the first RCT-designed study of an MHL educational programme for junior high school students in Japan. Only a few RCTs worldwide have investigated the effects of similar programmes for junior high school students, who belong to an age group that is characterized by a peak in the development of mental illness in a multidimensional manner, including actual behavioural changes [22].

First, the Sanita programme significantly improved the MIDUS scores immediately after the intervention and three months later, with a positive longitudinal effect on knowledge of mental illness. This positive result is consistent with previous studies [25]. In the 1st and 2nd classes, students learned about the incidence of mental illness, the common age of onset, the possibility of recovery, the necessity of early detection and appropriate management, and treatment and recovery from mental illness. In the 3rd class, they created a plan for how they could consult a professional if they experience mental health problems; this task seemed to have been effective for consolidating the knowledge acquired during the 1st and 2nd classes. The acquisition of appropriate knowledge about mental illness at the age of 12–13 years, which is close to the peak onset of mental illness, is very important from the viewpoint of primary prevention.

Secondly, the Sanita programme significantly improved the RIBS-J future scores immediately after the intervention. This result can be attributed to the fact that in the 1st class, the students acquired basic knowledge, such as the fact that anyone can develop a mental illness; as well, in the 3rd class, the students watched a video of a person with a mental illness and discussed the stigma of mental illness. Direct contact with people with mental illness has been reported to significantly improve attitudes and stigma towards people with mental illness [32]. Although the methods need to be further examined and sophisticated, since the present results showed no longitudinal effects of RIBS-J scores, methods such as the Sanita programme, which use video to create indirect contact with people with mental illness, are highly replicable in school settings. This is a very important finding for the dissemination of MHL educational programmes in the future. A typical programme developed in other countries is MindMatters in Australia [33]. This is a resource and professional development initiative that helps promote and protect the mental health, resilience, and social and emotional well-being of secondary school students. It consists of a variety of resources, including professional workshops for teachers and school-wide planning workshops for leaders and school teams. In addition, The Curriculum Guide, developed by Teen Mental Health in Canada, has instructional manuals, animations, and interviews available, which share some similarities with the Sanita programme [34]. Both programmes have extensive training content for teachers prior to conducting classes. The Sanita programme will need to incorporate pre-programme training for teachers as well.

Finally, there was no change in help-seeking behaviour (i.e., “consulting someone about mental illness”), which is the change that would have created the most benefit from this programme. A possible reason for this outcome is that the subjects of this study were not students who were already in need of some support, but rather a general sampling of public junior high school students. Most of the participants could have been healthy, and may have had less need for help-seeking behaviour in the short term [35]. These results may suggest that a follow-up period of longer than three months is desirable. There is also the possibility that this programme was not sufficient to promote behavioural change, and an examination of the factors and processes that lead to help-seeking behaviour might be needed to improve the effectiveness of MHL programmes. For adolescents, it may be useful to focus on more positive aspects, rather than just aiming to provide information on MHL. The previous psychoeducational interventions to improve self-efficacy, problem-solving, empathy, and coping strategies have been shown to promote psychological well-being [36,37]. In addition, the environment in which the young person lives is also very important, as well as the young person’s own factors in engaging help-seeking behaviour. In Japanese junior high schools, school nurses and counsellors are available to provide individual mental health counselling, but the rates of actual student use are extremely low [38]. Compared with other countries, there is, reportedly, a lack of people to turn to in times of trouble in Japan [39], and there is an urgent need to create a society in which people feel comfortable asking for help [40]. The usefulness of integrated (one-stop) services in the community as an early intervention approach to improve accessibility to help-seeking pathways for young people is increasingly being reported worldwide, and further research and practice is desirable [41,42].

Several limitations can be noted in the present study. First, because all measures were performed using self-report questionnaires, methodological limitations, including a social desirability bias, may have affected the findings. For example, mental-health-related information from the media or the personal experiences of the participants during the follow-up period may have improved the scores. However, as the RCT compared the effects of the interventions, these biases may not have had much impact on the results. Next, four new questions were used to investigate help-seeking behaviour, but these questions have not been adequately evaluated for reliability and validity. A standardised method for measuring actual help-seeking behaviour has not yet been established, and further investigation is needed. This study only included the participants from one junior high school where cooperation was obtained, and we may have been unable to accurately assess changes in help-seeking behaviour due to the small sample size. Although data are not available for junior high school students only, the annual incidence of mental illness estimated from a large epidemiological survey in Japan is approximately 4–6% [8]. If we attempt to recruit 100 people who develop mental illness in a 3-month follow-up period, we need to target more than 10,000 junior high school students. On the other hand, it is expected that students with more mild mental health problems, rather than with the onset of mental illnesses, would be much more prevalent. More epidemiological studies on the mental health of junior high school students are needed, and based on these studies, continued research is needed to examine the effectiveness of Sanita in a larger sample size. Although we used the RCT method to examine the effects of Sanita in this study, the methodology needs further refinement, especially to examine the effects in a larger number of samples. For example, using methods such as propensity score matching may be useful [43,44].

## 5. Conclusions

In conclusion, the Sanita MHL educational programme was longitudinally effective in improving junior high school students’ knowledge of mental illness, although the effects were insufficient to improve attitudes towards people with mental health problems and help-seeking behaviour during a short-term follow-up period. MHL education for young people, who are at the peak of the onset of mental illness, could be key for universal prevention strategies. Such education should be provided to all young people; this would require a society-wide approach, such as including it in the National Curriculum.

## Figures and Tables

**Figure 1 jpm-12-01281-f001:**
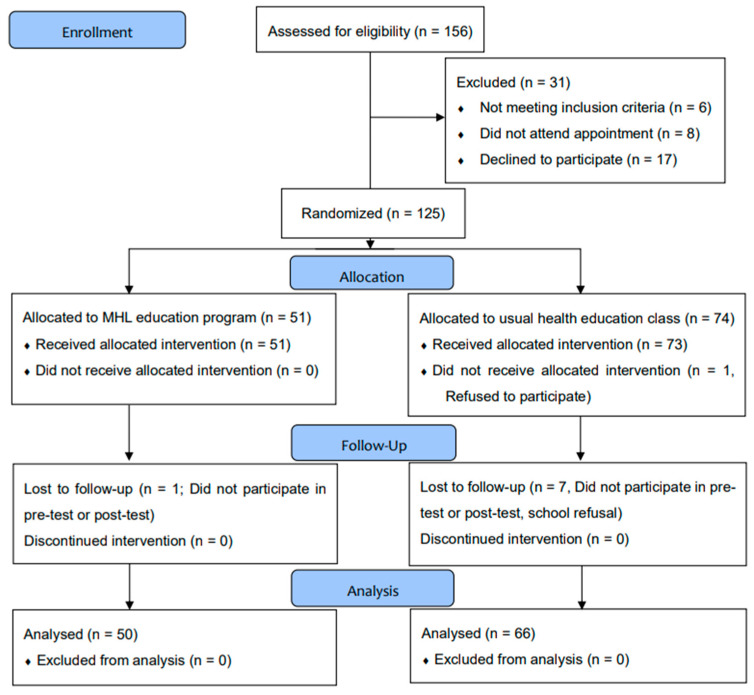
Flowchart of the survey.

**Table 1 jpm-12-01281-t001:** Contents of the “Sanita” mental health literacy educational programme.

Class	Aims and Learning Activities
1 (50 min)	Aims: Understand what mental illness is (including signs of mental health problems and characteristics of mental illness) Understand the prevalence of mental illness, the common age of its onset, the likelihood of recovery, and the need for early detection and appropriate treatment Understand how to prevent mental illness (including the importance, as with physical health, of practising a harmonious lifestyle, including appropriate exercise, diet, rest and sleep, and stress reduction) Learning Activities: i. Slide presentation on the characteristics of mental illness (10 min) ii. Slide presentation and discussion of common age of onset etc. (10 min) iii. Slide presentation on prevention, group discussion, and completion of worksheet (30 min)
2 (50 min)	Aims: Understand depression and schizophrenia and their main symptoms Understand anxiety and eating disorders and their main symptoms Understand the treatment of mental illness and the process of recovery (including the recognition of mental health problems and the main treatments) Learning Activities: i. Animation and slide presentation of typical symptoms of depression and schizophrenia (15 min) ii. Slide presentation on anxiety and eating disorders (10 min) iii. Group discussion and slide presentation of treatment options (25 min)
3 (50 min)	Aims: Identify people and resources to consult when they have a mental health problem Acquire specific methods of consulting when they have a mental health problem Acquire appropriate responses to prejudice and discrimination against mental illness Learning Activities: i. Brainstorming (15 min) ii. Slide presentation and exercise to develop a specific plan on how to consult (15 min) iii. Watching video interviews with people in remission from mental illness and discussing prejudice and discrimination (20 min)
Sanita [26].	

**Table 2 jpm-12-01281-t002:** Characteristics of participants in this study.

	MHL Group n = 50	Control Group n = 66	Statistical Values ^(a)^	
Sex, female, n (%)	25 (50.0)	36 (54.5)	χ^2^ = 0.24	*p* = 0.63
MIDUS, mean (SD)	19.40 (6.65)	21.00 (6.62)	t = −1.29	*p* = 0.20
RIBS-J future, mean (SD)	15.02 (3.24)	14.17 (3.09)	t = 1.44	*p* = 0.15
Behavioural experiences of				
“Talking to family”, n (%)	11 (22.0)	6 (9.1)	χ^2^ = 3.79	*p* = 0.05
“Visiting a website”, n (%)	6 (12.0)	8 (12.1)	χ^2^ = 0.00	*p* = 0.98
“Talking to a friend”, n (%)	6 (12.0)	5 (7.6)	χ^2^ = 0.65	*p* = 0.42
“Consulting someone”, n (%)	3 (6.0)	4 (6.1)	χ^2^ = 0.00	*p* = 0.99

MHL: Mental Health Literacy program; MIDUS: The Mental Illness and Disorder Understanding Scale; RIBS-J future: The Japanese version of the Reported and Intended Behaviour Scale, future part. ^(^^a)^ Group differences were tested using a *t*-test for continuous variables, and a chi-squared test for categorical variables.

**Table 3 jpm-12-01281-t003:** Effects of MHL group and control group on MIDUS and RIBS-J future scores.

	MHL Group Mean ± SD (Effect Size) ^(a)^	Control Group Mean ± SD (Effect Size) ^(a)^	Main Effect of Time	Main Effect of Group ^(^^b)^	Time × Group Interaction ^(^^b)^
			B [95% CI] *p* Value	B [95% CI] *p* Value	B [95% CI] *p* Value
MIDUS					
Baseline	19.40 ± 6.65	21.00 ± 6.62	0 (ref)	**1.60 [−9.65,12.85]** *p* = 0.780	0 (ref)
Post-test	11.00 ± 5.88 (1.34)	19.95 ± 7.34 (0.15)	**−8.40 [−10.07, −6.73]** ***p* < 0.001**		**7.36 [5.14,9.57]** ***p* < 0.001**
3-month follow-up	13.26 ± 5.80 (0.98)	19.05 ± 7.60 (0.27)	**−6.14 [−7.81, −4.47]** ***p* < 0.001**		**4.19 [1.97, 6.41]** ***p* < 0.001**
RIBS-J future					
Baseline	15.02 ± 3.24	14.17 ± 3.09	0 (ref)	**−0.85 [−5.97,4.26]** *p* = 0.743	0 (ref)
Post-test	17.22 ± 2.67 (0.74)	14.52 ± 3.35 (0.11)	**2.20 [1.45, 2.96]** ***p* < 0.001**		**−1.85 [−2.85, −0.85]** ***p* < 0.001**
3-month follow-up	16.86 ± 2.66 (0.62)	15.29 ± 3.21 (0.36)	**1.84 [1.85, 2.60]** ***p* < 0.001**		−0.72 [−1.72, 0.28] *p* = 0.158

Random-intercept and -slope (for time-point) linear regression mixed models (LRMM). B: Non-standardized regression coefficients; SD: standard deviation; CI: confidence interval; ref: reference; RIBS-J: Japanese version of Reported and Intended Behaviour Scale; MHL; Mental Health Literacy program; MIDUS, Mental Illness and Disorder Understanding Scale. ^(^^a)^ Cohen’s d, d = (M_Timet_ − M_Time0_)/SD_pooled,_ SD_pooled_ = √((SD_Timet_^2^ + SD_Time0_^2^)/2), time t = 1 (baseline survey), 2 (post-test survey), or 3 (3 follow-up survey); ^(^^b)^ reference group is the MHL group.

**Table 4 jpm-12-01281-t004:** Effects of MHL group and control group on behaviour scale.

	MHL Group Number (Proportion)	Control Group Number (Proportion)	Main Effect of Time	Main Effect of Group ^(a)^	Time × Group Interaction ^(a)^
			Odds Ratio [95% CI] *p* Value	Odds Ratio [95% CI] *p* Value	Odds Ratio [95% CI] *p* Value
“Talking to family”					
Baseline	11 (22.0)	6 (9.1)	0 (ref)	2.82 [0.05, 156.41] *p* = 0.612	0 (ref)
Post-test	21 (42.0)	11 (16.7)	**0.39 [0.19, 0.78]** ***p* = 0.008**		1.28 [0.43, 3.86] *p* = 0.656
3-month follow-up	13 (26.0)	12 (18.2)	0.80 [0.39, 1.67] *p* = 0.555		0.56 [0.18, 1.71] *p* = 0.309
“Visiting a website”					
Baseline	6 (12.0)	8 (12.1)	0 (ref)	0.99 [0.02, 42.37] *p* = 0.995	0 (ref)
Post-test	17 (34.0)	13 (19.7)	**0.27 [0.13, 0.53]** ***p* < 0.001**		2.12 [0.84, 5.39] *p* = 0.112
3-month follow-up	6 (12.0)	10 (15.2)	1.00 [0.46, 2.17] *p* = 0.100		0.77 [0.28, 2.11] *p* = 0.613
“Talking to a friend”					
Baseline	6 (12.0)	5 (7.6)	0 (ref)	1.66 [0.44, 63.26] *p* = 0.783	0 (ref)
Post-test	18 (36.0)	8 (12.1)	**0.24 [0.11, 0.52]** ***p*< 0.001**		2.45 [0.79, 7.63] *p* = 0.121
3-month follow-up	10 (20.0)	8 (12.1)	0.55 [0.25, 1.20] *p* = 0.132		1.09 [0.34, 3.46] *p* = 0.884
“Consulting someone”					
Baseline	3 (6.0)	4 (6.1)	0 (ref)	0.99 [0.06, 15.78] *p* = 0.994	0 (ref)
Post-test	4 (8.0)	7 (10.6)	0.73 [0.26, 2.10] *p* = 0.563		0.74 [0.19, 2.92] *p* = 0.667
3-month follow-up	2 (4.0)	4 (6.1)	1.53 [0.44, 5.35] *p* = 0.502		0.65 [0.14, 3.17] *p* = 0.595

Random-intercept and -slope (for time-point) logistic regression mixed models. ^(^^a)^ Reference group is the MHL group.

## Data Availability

The data that support the findings of this study are available upon reasonable request to the corresponding author. The data are not publicly available due to privacy and ethical restrictions.

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
