# Peer review of "Effectiveness of a Comprehensive Mental Health Literacy Educational Programme for Junior High School Students: A Randomised Controlled Trial Examining Changes in Their Knowledge, Attitudes, and Behaviour"

_jpm, 2022, doi:10.3390/jpm12081281_

Round 1
Reviewer 1 Report
This paper study program in junior high school students. Using RCT to evaluate MHL effectiveness, qiestionnaries before, after, and three months after the program was recorded to compare the effectiveness of the MHL program.
The study is important for students mental health and understand behavioral change. Author claimed it is the first application using RCT in this kind. However the novelty in the methodology is not high.
There are some suggestions for the authors to consider.
The improvement in help-seeking behavior was not observed significantly. This may be a result of insufficient subjects population. I would suggest the authors continue this study on more students that potentially have the target disorder risk. The minimum requirement of subjects size should be discussed base on the averaged percentage of mental disorders among junior high students in Japan.
The literature review of RCT and alternative methods could be further strengthened. Some of the references related to the methodology are outdated, more recent developments in this field should be analyzed and compared.
Author Response
Point-by-point responses
Reviewer #1
This paper study program in junior high school students. Using RCT to evaluate MHL effectiveness, qiestionnaries before, after, and three months after the program was recorded to compare the effectiveness of the MHL program. The study is important for students mental health and understand behavioral change. Author claimed it is the first application using RCT in this kind. However the novelty in the methodology is not high. There are some suggestions for the authors to consider.
Reply: Thank you for the positive feedback. We are grateful to Reviewer #1 for the useful suggestions that have helped us improve our paper.
- The improvement in help-seeking behavior was not observed significantly. This may be a result of insufficient subjects population. I would suggest the authors continue this study on more students that potentially have the target disorder risk. The minimum requirement of subjects size should be discussed base on the averaged percentage of mental disorders among junior high students in Japan.
Reply: We thank the reviewer for the insightful suggestion. In response to your suggestion, we have added the sentences regarding the sample size in the Discussion, as follows:
(Page 9, lines 281-291) “This study only included the participants from one junior high school where cooperation was obtained, and we may have been unable to accurately assess changes in help-seeking behavior due to the small sample size. Although data are not available for junior high school students only, the annual incidence of mental illness estimated from a large epidemiological survey in Japan is approximately 4-6% [8]. If we attempt to recruit 100 people who develop mental illness in a 3-month follow-up period, we need to target more than 10,000 junior high school students. On the other hand, it is expected that students with more mild mental health problems, rather than the onset of mental illnesses, would be much more prevalent. More epidemiological studies on the mental health of junior high school students are needed, and based on these studies, continued research is needed to examine the effectiveness of Sanita in a larger sample size.”
Added reference:
- Nishi, D.; Ishikawa, H.; & Kawakami, N. Prevalence of mental disorders and mental health service use in Japan. Psychiatry and Clinical Neurosciences, 2019, 73(8), 458–465. https://doi.org/10.1111/pcn.12894
- The literature review of RCT and alternative methods could be further strengthened. Some of the references related to the methodology are outdated, more recent developments in this field should be analyzed and compared.
Reply: As suggested by the reviewer, we have added a sentence to the very end of the Discussion section, as follows:
(Page 9-10, lines 292-295) “Although we used the RCT method to examine the effects of Sanita in this study, the methodology needs further refinement, especially to examine these in a larger number of samples. For example, using methods such as propensity score matching may be useful [42, 43].”
Added references:
42 Demetrios, K.; Roger, L. Confounding by Indication in Clinical Research. JAMA. 2016, 316(17), 1818-1819. doi:10.1001/jama.2016.16435
43 Issa, D.; Radley, S.; Jessica, P.; Mei, C.; Vasileia, V.; Haseeb, J.; Jeremy, R.; Thomas, T.; Georgios, K. Do observational studies using propensity score methods agree with randomized trials? A systematic comparison of studies on acute coronary syndromes. European Heart Journal, 2012, 33, 1893–1901. https://doi.org/10.1093/eurheartj/ehs114

Reviewer 2 Report
An interesting study on the prevention programme and mental health knowledge. Methodologically sound. Statistical analysis correct. Discussion scientifically maturely led. I would add information about similar projects in other countries to the discussion.The work can be accepted in its current form.
Author Response
Point-by-point responses
Reviewer #2
- An interesting study on the prevention programme and mental health knowledge. Methodologically sound. Statistical analysis correct. Discussion scientifically maturely led. I would add information about similar projects in other countries to the discussion. The work can be accepted in its current form.
Reply: Thank you for the positive feedback. We are grateful to Reviewer #2 for the useful suggestions that have helped us improve our paper. We fully agree with the reviewer’s comments. We have added sentences regarding the programs in Australia and Canada in the Discussion, as follows:
(Page 8-9, lines 238-248) “A typical programme developed in other countries is MindMatters in Australia [32]. This is a resource and professional development initiative that helps promote and protect the mental health, resilience, and social and emotional well-being of secondary school students. It consists of a variety of resources, including professional workshops for teachers and school-wide planning workshops for leaders and school teams. In addition, The Curriculum Guide, developed by Teen Mental Health in Canada, has instructional manuals, animations, and interviews available, which share some similarities with the Sanita programme. Both programmes have extensive training content for teachers prior to conducting classes. The Sanita programme will need to incorporate pre-programme training for teachers as well.”
Added references:
32 Australian Government Department of Health, MindMatters. Available online: https://www1.health.gov.au/internet/publications/publishing.nsf/Content/suicide-prevention-activities-evaluation~positioning-the-nspp~mindmatters(accessed on 23 July 2022)
33.TeenMentalHealth.org. School Mental Health The Curriculum Guide. Available online: https://mhlcurriculum.org/about-the-guide/download-the-guide/(accessed on 23 July 2022)
